# A Prospective Observational Study Analyzing the Diagnostic Value of Hepcidin-25 for Anemia in Patients with Inflammatory Bowel Diseases

**DOI:** 10.3390/ijms25073564

**Published:** 2024-03-22

**Authors:** Stanko Petrović, Dino Tarabar, Danica Ćujić, Dusica Stamenkovic, Marijana Petrović, Nemanja Rančić, Vesna Subota, Nenad Perišić, Mihailo Bezmarević

**Affiliations:** 1Medical Faculty of the Military Medical Academy, University of Defence, 11 000 Belgrade, Serbia; dusicastamenkovic@yahoo.com (D.S.); makystan@yahoo.com (M.P.); gastronesa1@gmail.com (N.P.); bezmarevicm@gmail.com (M.B.); 2Clinic for Gastroenterology and Hepatology, Military Medical Academy, University of Defence, 11 000 Belgrade, Serbia; 3Clinic for Gastroenterology, University Hospital Center “Dr. Dragiša Mišović”, 11 050 Belgrade, Serbia; dino.tarabar@gmail.com; 4Institute for Application of Nuclear Energy, University of Belgrade, 11 000 Belgrade, Serbia; danicac@inep.co.rs; 5Department of Anesthesiology and Intensive Care, Military Medical Academy, 11 000 Belgrade, Serbia; 6Clinic for Nephrology, Military Medical Academy, 11 050 Belgrade, Serbia; 7Centre for Clinical Pharmacology, Military Medical Academy, University of Defence, 11 000 Belgrade, Serbia; nece84@hotmail.com; 8Institute of Medical Biochemistry, Military Medical Academy, University of Defence, 11 000 Belgrade, Serbia; subota.vesna@gmail.com; 9Unit for Perioperative Nutrition, Department of Hepatobiliary and Pancreatic Surgery, Clinic for General Surgery, Military Medical Academy, University of Defence, 11000 Belgrade, Serbia

**Keywords:** hepcidin, ferritin, ulcerative colitis, Crohn’s disease, anemia, inflammation, cytokine

## Abstract

Iron deficiency (IDA) and chronic disease (ACD) anemia are complications of inflammatory bowel diseases (IBDs). Therapeutic modalities in remission and active IBD depend on the type of anemia. This study evaluated the link between hepcidin-25, proinflammatory cytokines, and platelet activation markers as biomarkers of anemia and inflammation in active IBD and remission. This prospective observational study included 62 patients with IBD (49 with ulcerative colitis and 13 with Crohn’s) and anemia. Patients were divided into Group I (no or minimal endoscopic signs of disease activity and IDA), Group II (moderate and major endoscopic signs of disease activity and mild ACD), and Control group (10 patients with IBD in remission, without anemia). We assessed the difference among groups in the levels of CRP, hemoglobin (Hgb), serum iron, ferritin, hepcidin-25, interleukins, TNF–α, IFN-γ, soluble CD40 ligand, and sP-selectin. Hepcidin-25 levels were significantly higher in Group II versus Group I (11.93 vs. 4.48 ng/mL, *p* < 0.001). Ferritin and CRP values showed similar patterns in IBD patients: significantly higher levels were observed in Group II (47.5 ng/mL and 13.68 mg/L) than in Group I (11.0 ng/mL and 3.39 mg/L) (*p* < 0.001). In Group II, hepcidin-25 was positively correlated with ferritin (ρ = 0.725, *p* < 0.001) and CRP (ρ = 0.502, *p* = 0.003). Ferritin was an independent variable influencing hepcidin-25 concentration in IBD patients, regardless of disease activity and severity of anemia. IBD hepcidin-25 best correlates with ferritin, and both parameters reflected inflammation extent and IBD activity.

## 1. Introduction

Anemia is one of the most common complications in inflammatory bowel disease (IBD), with a prevalence of 27% in Crohn’s disease (CD) and 21% in ulcerative colitis (UC) [1]. The most common types of anemia in IBD are iron deficiency anemia (IDA), seen in over 50% of patients, and chronic disease anemia (ACD) [2]. Other types of anemia might also be present in IBD patients, including vitamin B12 or folic acid deficiency anemia, hemolytic anemia, or drug-induced anemia [2]. The causes of ACD are iron capture in macrophages of the reticuloendothelial system, shortened erythrocyte lifespan, and myelosuppression. Additionally, proinflammatory cytokines (IL-1, IL-6, TNF-α) mediate ACD [1]. To date, no reliable biomarker can clearly distinguish IDA from ACD.

To identify the cause of anemia, multiple laboratory tests are used in practice: ferritin, transferrin, transferrin saturation, mean corpuscular volume (MCV), red cell distribution width (RDW), soluble transferrin receptor (sTFR), sTFR/Log ferritin, hepcidin-25, and others [2]. Hepcidin-25 is a 25-amino-acid-long peptide, coded by the *HAMP* gene, that plays a central role in iron homeostasis [3,4]. Hepcidin-25 acts by binding to ferroportin and leading to its internalization and lysosomal degradation, thus preventing iron release from enterocytes and macrophages in the liver and spleen [5]. Shorter hepcidin variants with 20 or 22 amino acids are products of hepcidin-25 degradation and are biologically less active. The antimicrobial activity of hepcidin, similar to that of defensins, was demonstrated in vitro [4]. The liver is the primary site of hepcidin-25 synthesis, but other organs (brain, gut, stomach, pancreas, heart, lungs) may synthesize hepcidin with a possible role in the immune response [6]. The central stimuli for hepcidin-25 synthesis are iron overload and inflammation, while iron deficiency, hypoxia, and increased erythropoiesis cause reduced synthesis of hepcidin-25 [7]. Hepcidin-25 secretion is induced by interleukin (IL-6) in inflammatory states [8]. However, an iron overload induces its secretion by bone morphogenetic proteins 6 and hemojuvelin, while matriptase-2 prevents overexpression of hepcidin-25 [7]. In IBD patients, hepcidin-25 has been shown to correlate with other markers of inflammation: ferritin, CRP, and IL-6 [9,10]. 

Inflammation in IBD might be often amplified by activated platelets [11,12]. The link between hepcidin-25 and inflammatory markers, including cytokines and platelet activation parameters, such as sCD40L and sP-selectin, has not yet been clearly revealed [11]. Limited knowledge exists regarding the role of hepcidin-25 in the active phase and remission of IBD, as well as the relation between hepcidin-25 and CRP or IL-6 in different IBD phases [13]. 

The main aim of this study was to assess serum hepcidin-25 in silent and active IBD, as well as to explore the relationship between hepcidin-25 and markers of inflammation. 

## 2. Results

### 2.1. Patients Characteristics

There was no difference in the demographic characteristics of patients between subjects in both groups (Appendix A). The distribution of patients according to disease location, extension, and therapy is summarized in Table 1.

### 2.2. Parameters of Iron Status and Inflammation in IBD Patients

The hepcidin-25 values were significantly higher in Group II than in the Control group [11.93 (IQR 4.82–36.30) ng/mL vs. 5.30 (IQR 4.58–8.41) ng/mL] (Appendix A). However, Group I had lower hepcidin-25 values compared to the Control group [4.48 (IQR 4.20–4.79) ng/mL vs. 5.30 (IQR 4.58–8.41) ng/mL] (Appendix A).

The hepcidin-25 levels were significantly lower in Group I compared to Group II [4.48 (4.20–4.79) ng/mL vs. 11.93 (4.82–36.30) ng/mL, *p* < 0.001] (Figure 1A). A similar finding was observed in both ulcerative colitis (UC) subgroups [4.34 (4.19–4.78) ng/mL vs. 12.28 (4.70–40.75) ng/mL, *p* < 0.001] (Figure 1A). Mean Hgb levels were also lower in patients in Group I compared with Group II [110.0 ± 23.4 g/L vs. 126 ± 21.2 g/L, *p* < 0.05] (Figure 1B). A similar Hgb change pattern was seen in UC subgroups, lower in UC Group I compared to UC Group II [108.0 ± 20.3 g/L vs. 128.0 ± 20.2 g/L *p* < 0.01] (Figure 1B). Accordingly, the values of ferritin and CRP, inflammatory indicators, expressed the same distribution pattern of values. Ferritin levels in Group I (11.00 (5.61–24.02) ng/mL, as well as in subgroup I (9.98 (5.29–16.60) ng/mL) were lower than in Group II [47.50 (14.38–90.37) ng/mL, (*p* < 0.001)] and subgroup II [52.15 (16.74–91.52) ng/mL, (*p* < 0.001)] (Figure 1C). The CRP values were higher in Group II than in Group I [3.39 (1.30–6.05) mg/L vs. 13.68 (4.41–35.23) mg/L, *p* < 0.001]. A similar difference was observed in UC patients vs. UC II [13.68 (4.59–39.37) mg/L vs. 3.50 (2.68–6.04) mg/L, *p* < 0.001] (Figure 1D). Notably, serum iron levels were low in most patients, and no differences in serum iron levels existed between groups (Figure 1E). There was a strong positive correlation between iron concentration and hepcidin in CD Group I (r = 0.775, *p* = 0.041).

Since levels of hepcidin-25, ferritin, and CRP were markedly higher in patients with active disease than in those in remission, positive correlations between hepcidin-25 and ferritin (Spearman’s ρ 0.725, *p* < 0.001), as well between hepcidin-25 and CRP (Spearman’s ρ 0.502, *p* = 0.003), were found in Group II (Figure 2).

In CD patients in Group I, a strong positive correlation was shown between hepcidin-25 and iron (Spearman’s ρ 0.775, *p* = 0.0041), as well as between hepcidin-25 and IL-6 (Spearman’s ρ 0.894, *p* = 0.007). Given the small number of patients with CD, this finding should be interpreted cautiously. Univariate analysis showed ferritin and hemoglobin as the variables with the highest positive influence on the hepcidin-25 concentration in Group I. In contrast, in Group II, the main predictors of hepcidin-25 concentration were ferritin and CRP. Multivariate analysis found that only ferritin was an independent predictor of hepcidin-25 levels in both IBD patient groups (Table 2). These results lead to the conclusion that the serum level of ferritin has a significant influence on hepcidin-25 synthesis: when serum levels of ferritin are low, synthesis of hepcidin-25 is inhibited, while increased ferritin level causes a rise in hepcidin-25 synthesis.

To estimate the diagnostic utility of hepcidin-25 in assessing IBD disease severity, receiver operating characteristic (ROC) curve analysis was performed. Based on ROC curve analysis (Figure 3), hepcidin-25 was a good predictor of disease activity in IBD, classifying patients into Groups I or II as expected. At the cut-off value of 4.79, sensitivity was 78.1% and specificity was 76.7% for patients belonging to Group II. ROC curve analysis (AUC = 0.749) supported hepcidin-25 as a valuable biomarker for the assessment of inflammation activity in IBD patients and an indicator of whether the patient will be in the first group (anemia) or in the second group (gut inflammation is dominant) (Figure 3).

### 2.3. Cytokine Profile in IBD Patients

A panel of 13 mostly proinflammatory cytokines was also performed in each patient (Appendix A). A significant difference in measured levels between Group I and Group II was observed solely for cytokine IL-13 (Appendix A). Levels of IL-13 were higher in patients in Group I compared to those in Group II [287.50 (97.25–440.25) pg/mL vs. 132.50 (19.00–211.75) pg/mL, *p* < 0.01] (Appendix A). The same was seen in patients with ulcerative colitis UC I and II subgroups [293.00 (125.00–440.00) pg/mL vs. 132.50 (16.00–234.20) pg/mL] (Appendix A). No relevant differences in concentrations of other cytokines, sCD40L, or sP-selectin were found between the groups (Appendix A).

## 3. Discussion

Anemia is a frequent comorbidity in IBD patients. The true cause of anemia and its treatment is a common issue in clinical practice. Serum ferritin is often used to distinguish IDA from other types of anemia [14]. Ferritin levels below 30 ng/mL, regardless of inflammation, are indicative of IDA. Serum ferritin levels ranging from 30 to 100 ng/mL, in parallel with elevated parameters of inflammation, suggest mixed ACD and IDA, while if ferritin is above 100 ng/mL, ACD alone is likely [15]. Yet, the diagnostic accuracy of ferritin in the differential diagnosis of anemia in IBD is limited. Ferritin, an acute-phase protein, might be elevated in patients with active IBD even if iron body reserves are low. So, additional markers are needed, and we focused here on hepcidin-25 as the main regulator of iron homeostasis.

In this pilot study, IBD patients were divided into two groups based on the severity of anemia and mucosal inflammation. Serum hepcidin-25 levels were lower in IBD patients with sideropenic anemia (Group I) compared to patients with active inflammation in the gastrointestinal tract (Group II). Since hepcidin-25 serum levels are mainly affected by inflammation and iron reserves in the body [5,6], this finding suggests that proinflammatory mediators probably stimulate hepatic hepcidin-25 secretion during the active IBD phase. On the other hand, patients with severe IDA, regardless of inflammation, had low hepcidin-25 levels, usually close to the values seen in a healthy population or even lower. Several previous studies had similar findings [7,10,16], suggesting that in pronounced IDA, anemia-driven signals predominate inflammation as a regulatory signal in hepcidin-25 expression. In contrast to our findings, Arnold and coworkers found lower serum hepcidin-25 levels in IBD patients both with or without IDA compared to a healthy population [10]. 

As expected, patients with active IBD (Group II) also had elevated ferritin and CRP levels compared to patients with silent or mild IBD (Group I). In patients within Group II and in a subgroup of patients with active UC (UC Group II), a positive correlation between hepcidin-25 and both acute-phase proteins, ferritin and CRP, was found. Other authors obtained similar results [9,17,18,19]. A positive link between serum hepcidin-25 and disease severity was also found in other inflammatory conditions, such as acute pancreatitis, coronary heart disease, rheumatoid arthritis, and malignancy [20,21,22]. We assumed that severe inflammation led to increased secretion of both hepcidin-25 and ferritin from the liver. A strong positive correlation between hepcidin-25 and IL-6 levels, found in a subgroup of patients with active CD, is in line with previous reports [10,19]. 

While low serum hepcidin-25 levels are common in IDA patients, in patients with ACD, hepcidin-25 levels might be already elevated. In Group I patients, where anemia predominates and IBD is silent, ferritin and hemoglobin were found to be the most relevant predictors for hepcidin-25 concentration. Among patients with active inflammation (Group II), hepcidin-25 levels were mainly influenced by ferritin and CRP. Therefore, in IBD, regardless of disease activity, ferritin could be the most important predictor of hepcidin-25 serum levels. Oustamanolakis et al. found significantly higher hepcidin in patients with UC and patients with CD compared with healthy controls, and the correlation between hepcidin and disease activity and ferritin was the same as in our study [17]. Yet, the results of different studies are not consistent, and Paköz et al. did not find any difference in hepcidin-25 levels between patients with active IBD and those in remission; neither detected a link between hepcidin-25 and inflammatory markers CRP and IL-6 in IBD patients [13]. There is a possibility that hepcidin-25 serum concentrations decrease at a certain degree in anemia, regardless of the grade of inflammation [11]. This point could be critical, both in anemia development in IBD and also for treatment intervention.

In order to evaluate the diagnostic utility of hepcidin-25 in assessing IBD activity, an ROC analysis was performed. A cut-off value for hepcidin-25 was set at 5 ng/mL, meaning that IBD patients with serum hepcidin-25 above 5 ng/mL have an 80% higher probability for mucosal inflammation and ACD, with or without IDA. If anemia is not clinically significant, serum hepcidin-25 levels below 5 ng/mL suggest mucosal healing in IBD patients. However, if anemia is pronounced (Hgb levels < 100 g/L), serum hepcidin-25 will remain low regardless of the IBD activity. These results are supported by the fact that anemia inhibits hepcidin-25 synthesis [23]. Our results are justified by the hepcidin-25 levels: Group I patients (with anemia and the absence of inflammation) had lower levels of serum hepcidin-25 compared to Control group and Group II patients (predominant inflammation, minimal anemia).

Although IL-6 induces hepcidin-25 synthesis in the liver via STAT3 [24], it seems that this signal is not essential under inflammation. In animal studies, *HAMP* gene expression was also increased in IL-6-deficient mice with chronic inflammation [11], suggesting that other cytokines might affect hepcidin-25 secretion. Shu and coauthors found a positive correlation between serum hepcidin-25 and IL-6, TNF-α, and IL-17, but not between hepcidin-25 and IFN-γ [25]. Anti-TNF-α therapy could improve anemia in IBD patients by downregulating hepcidin expression and normalizing iron metabolism [25,26]. Our group also had positive anti-TNF-α effects in regulating anemia severity as well. In addition, platelets, endothelium, and T-cells express proinflammatory activity and participate in the development, maintenance, and amplification of inflammation in IBD. Therefore, we explored serum concentrations of cytokines secreted by cells of innate or acquired immunity and determined serum values of sCD40L and sP-selectin and markers of platelets, endothelium, and T-lymphocyte activation. However, none of the 13 cytokines investigated here, nor sP selectin nor sCD40L, were correlated with the serum hepcidin-25. 

The findings of this study provide an overview of the anemia profile in patients with IBD. An imbalance of iron homeostasis is the basis of anemia in IBD, but anemia cannot be simply corrected by iron supplementation. Iron homeostasis, regulated by different factors at the molecular, cellular, and systemic levels, and iron intestinal absorption and storage/mobilization are often impaired in pathophysiological conditions [27]. This is particularly expressed in IBD, where inflammation and blood loss are present. Initially, inflammation of the intestinal wall induces hepatic hepcidin-25 secretion, and serum levels rise above those seen in the healthy population. With the development of mucosal erosion and ulcers, bleeding leads to a hemoglobin decrease, and anemia (IDA, ACD, or both) occurs. At the point when hemoglobin levels drop below the critical level, anemia becomes the dominant signal that governs hepcidin-25 synthesis. Now, hepcidin-25 levels fall, sometimes even below the serum levels seen in the healthy population, regardless of inflammation presence and severity. At this point, anemia correction in IBD should be conducted. Otherwise, correction of mild anemia under active inflammation would hardly be effective, as elevated hepcidin-25 would prevent iron utilization. Oral iron preparations would probably be ineffective under inflammation and mild anemia, as elevated hepcidin-25 would prevent intestinal iron absorption. In addition, under high hepcidin-25 levels, intravenous administrated iron would be directed and stored in macrophages instead of hematopoietic tissues. Therefore, determining hepcidin-25 concentration in the serum of IBD patients could be very useful for the assessment of anemia severity and treatment decisions, particularly during prolonged periods of active disease, when inflammation persists and severe anemia might develop. 

Our findings need exploration in more extensive trials since they can significantly change how we treat anemia in IBD. There are several limitations of this study, including the relatively small number of patients and the overall heterogeneity of IBD patients. Although we analyzed subgroups with CD and UC, the sample is too small to exclude bias. Unfortunately, we were not able to measure sTFR and transferrin or to determine fecal calprotectin in IBD patients.

## 4. Materials and Methods

### 4.1. Study Design

This prospective observational study included patients with a diagnosis of IBD treated in our hospital during the period from 2016 to 2018. This study was approved by the Ethics Committee of the Military Medical Academy (No. 20/2021, 24 April 2021) in accordance with the Helsinki Declaration.

### 4.2. Patients and Definitions

This study included 62 patients with IBD (49 with UC and 13 with CD) and concomitant anemia (Figure 4). According to the World Health Organization, anemia is defined as hemoglobin concentration below 130 g/L in men over 15 years of age and below 120 g/L in non-pregnant women over 15 years of age [28].

IDA was defined as ferritin below 30 ng/mL with concurrent anemia, and ACD was defined as ferritin between 30 and 100 ng/mL with anemia and increased CRP [29]. Mayo endoscopic subscore was used for assessing endoscopic disease activity for UC [30], and the Simple Endoscopic Score was used for CD [31]. Severity of anemia was graded based on hemoglobin as mild (110–129 g/L), moderate (80–109 g/L), and severe (<80 g/L) [32].

All study participants underwent colonoscopy with a colonic mucosa biopsy, while in patients with CD, a terminal ileum biopsy was also performed. Endoscopic signs of ulcerative colitis (UC) activity included minimal signs—Mayo 0 or 1 (erythema, decreased vascular pattern, and mild friability), moderate signs—Mayo 2 (absent vascular pattern and erosions), and severe signs—Mayo 3 (spontaneous bleeding and ulcerations) [30]. CD score was calculated for each colon and terminal ileum segment depending on the ulcer’s size, ulcerated surface, affected surface, and presence of narrowing [31].

Patients were divided into 2 groups depending on the activity of the IBD and the severity of anemia (Figure 4). Group I included patients with no endoscopic signs of disease activity, or minimal signs, mainly with IDA and symptoms of anemia [33,34]. Group II included patients with moderate and major endoscopic signs of disease activity and mild ACD with or without IDA, who complained mainly of diarrhea [33,34]. The Control group consisted of 10 IBD patients in remission, without anemia (Figure 4). 

Inclusion criteria: patients with IBD and anemia.

Exclusion criteria: age under 18 years, concomitant or previously diagnosed malignant disease, heart failure, liver cirrhosis, chronic obstructive pulmonary disease, hemoglobinopathies, autoimmune hemolytic anemia, myelodysplastic syndrome, chronic kidney disease (glomerular filtration rate < 30 mL/min/1.73 m^2^), previous (6 months before entering this study) or current treatment with iron supplementation and/or erythropoietin (EPO), pregnant or lactating women, patients with active infection, presence of any extra-intestinal manifestation of IBD, and the patient’s desire not to participate in the study. Also, other causes of anemia (hemolysis, EPO/vitamin B12/folic acid deficiency, gluten-sensitive enteropathy, and renal failure) were excluded.

### 4.3. Sample Size Calculation

Based on the standard statistical parameters (study power—0.90; α error probability—0.05; two-tailed testing; equal group sizes) to find a significant difference in hepcidin-25 levels between the anemic IBD group and the IBD group that did not have anemia (according to the literature, the level of hepcidin-25 in the IBD group without anemia is 6.81 ± 1.2 ng/mL, i.e., in the IBD group with anemia, it is 4.14 ± 0.72 ng/mL [10]), the obtained effect size was 2.6982088. Applying these parameters, we calculated the required sample size of 10 patients with a t-test for independent samples, using the computer program G*Power 3.1. Therefore, 10 patients were included in the Control group. In comparison, 30 patients with IBD, with or without anemia, were included in the other two groups to ensure sufficient group sizes for analyses.

### 4.4. Laboratory Parameters

The levels of hepcidin-25, hemoglobin (Hgb), serum iron, ferritin, CRP, cytokine panel including 13 cytokines (interleukins 2, 4, 5, 6, 9, 10, 13, 17A, 17F, 21, 22, TNF-α, IFN-γ), soluble CD40 ligand (sCD40L), and soluble P-selectin (sP-selectin) were determined in all patients. To exclude other causes of anemia, additional parameters were analyzed: haptoglobin, lactate dehydrogenase (LDH), creatinine, erythropoietin (EPO), vitamin B12, folic acid, and anti-tissue transglutaminase antibodies (both IgG and IgA classes). Venous blood samples from patients with IBD were collected on the day when colonoscopy was performed. Hemoglobin levels were measured from EDTA blood samples immediately after sample collection on an automated hematology analyzer (ADVIA 1800, Dimension RxL Max, Siemens, Germany). For biochemical and immunochemical tests, blood samples were collected in serum tubes, without additives, centrifuged at 2000× *g* for 10 min at room temperature, aliquoted and analyzed on the same day, or stored at −80 °C until the moment of test performance. Concentrations of biochemical and some immunological parameters were determined using routine tests on an automated analyzer (ADVIA 1800, Dimension RxL Max, Siemens, Germany). 

Quantitative measurement of hepcidin-25 in human serum was performed using Hepcidin-25 Chemiluminescent Direct ELISA assay (Corgenix Inc., Broomfield, CO, USA). The test principle is sandwich ELISA: in the first step, hepcidin from calibrators, controls, and samples was captured by a mouse anti-hepcidin 25 antibody immobilized on a microtiter plate. Following the incubation and washing step, a detection antibody (mouse anti-hepcidin 25 antibody conjugated to peroxidase) was added. After the second incubation and washing step, an enzyme substrate was added, and a chemiluminescent signal was measured on a multilabel reader, Victor 3V (Perkin Elmer Inc., Waltham, MA, USA). The measured signal was directly proportional to the levels of hepcidin-25 and, the concentration was calculated based on a calibration curve (range: 0–203 ng/mL).

Concentrations of sP-selectin and sCD40L were determined using commercial ELISA tests for Human sSELP and Human sCD40L (Elabscience Biotechnology Co. Ltd., Wuhan, China). ELISA tests were performed according to the manufacturer’s instructions. sCD40L ELISA kit No E-EL-H0035, Detection Range: 62.5–4000 pg/mL, Sensitivity: 37.50 pg/mL. pSelectin ELISA kit No E-EL-H 6180, Detection Range: 31.25–2000 pg/mL, Sensitivity: 18.75 pg/mL. 

To avoid inter-assay variability, all samples were tested at once for a particular analyte. The reproducibility of ELISA tests expressed as the coefficient of variation was less than 10%.

Serum levels of cytokines (IL-2, IL-4, IL-5, IL-6, IL-9, IL-10, IL-13, IL-17A, IL-17F, IL-21, IL-22, TNF-α, IFN-γ) were measured using a commercial flow cytofluorometry kit (Human Th1/Th2/Th9/Th17/Th22 BIOLEGEND 13 plex Kit) on a Beckman Coulter FC500 flow cytometer.

### 4.5. Statistical Analysis

Statistical analysis was conducted using the IBM SPSS Statistics version 26.0 program (Chicago, IL, USA) and GraphPad Prism Software version 5 (GraphPad Software Inc., San Diego, CA, USA). The Shapiro–Wilk test was used to test the normality of data distribution because the number of subjects was <50. Except for Hgb, all other parameters showed a non-Gaussian distribution. Results are presented as mean ± standard deviation or median with interquartile range (25th–75th percentile) depending on the data type and distribution. 

Differences between groups were tested either with ANOVA and Bonferroni Multiple Comparison post hoc test (for normal distributions) or with Kruskal–Wallis followed by Mann–Whitney U (for non-Gaussian distributions). The correlation analysis was performed using the Spearman correlation. The association between parameters was calculated by univariate and multivariate linear regression. Calculations of odds ratios and their 95% confidence intervals were performed to determine the strength of the association between hepcidin-25 and other variables, the IBD activity, and the severity of anemia. Binary logistic regression analyses incorporated the most promising independent variables as a single risk factor (unadjusted) and taken together (adjusted). 

Receiver operating characteristic (ROC) curve analysis investigated hepcidin-25 and other variables as predictors of the activity of the IBD and the severity of anemia. The cut-off value for hepcidin-25 was calculated according to the ROC curve. The sensitivity and specificity of hepcidin-25 and other variables for the activity of IBD and the severity of anemia were described as cut-off values with a 95% confidence interval (CI). The p-values less than 0.05 were considered statistically significant.

## 5. Conclusions

Based on the obtained results, the best correlation was found between hepcidin-25 and ferritin. However, based on previously mentioned ferritin limitations, hepcidin-25, a significant regulator of iron homeostasis, might better indicate iron body status, particularly under active inflammation. Given the excellent correlation between ferritin and hepcidin-25, as demonstrated here, the determination of ferritin might be proposed for the assessment of anemia in IBD patients. 

The methodological restrictions are currently the main limitation for using hepcidin-25 in routine laboratory practice. In the future, determining hepcidin-25 will be of utmost importance in patients with intense inflammation and advanced anemia, as they will not benefit from iron supplementation until the inflammation is resolved and hepcidin-25 decreases to normal levels. For now, determining hepcidin-25 could be an essential parameter in research and resolving the complex etiopathogenesis background of anemia in IBD patients.

## Figures and Tables

**Figure 1 ijms-25-03564-f001:**
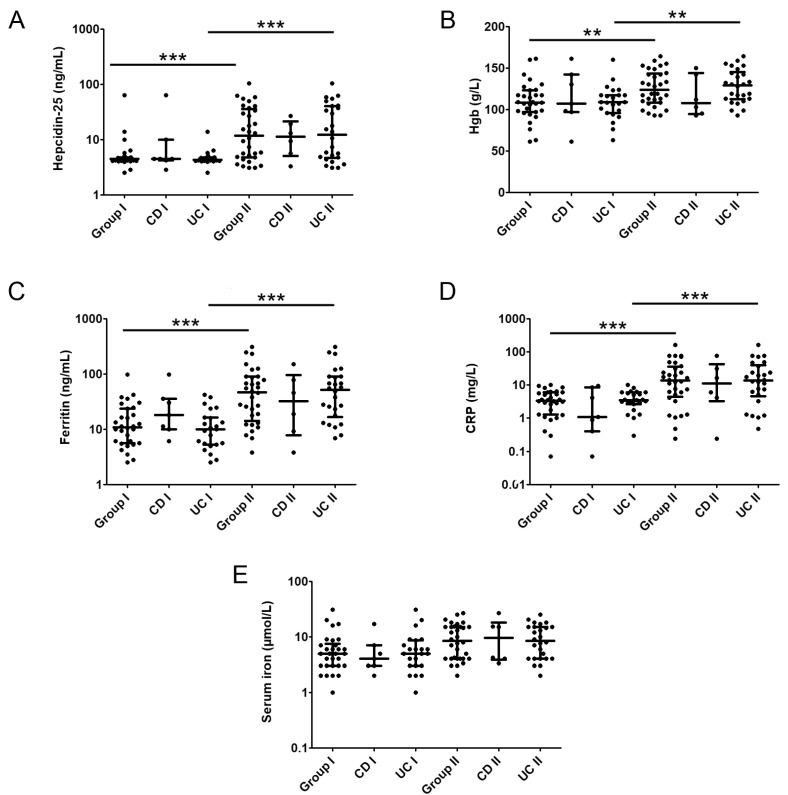
Serum levels of hepcidin-25 (**A**), Hgb (**B**), ferritin (**C**), CRP (**D**), and serum iron (**E**) in IBD patients in Group I and Group II. Within groups, patients are further divided into ulcerative colitis (UC) and Chron’s disease (CD) subgroups; **—*p* < 0.01, ***—*p* < 0.001.

**Figure 2 ijms-25-03564-f002:**
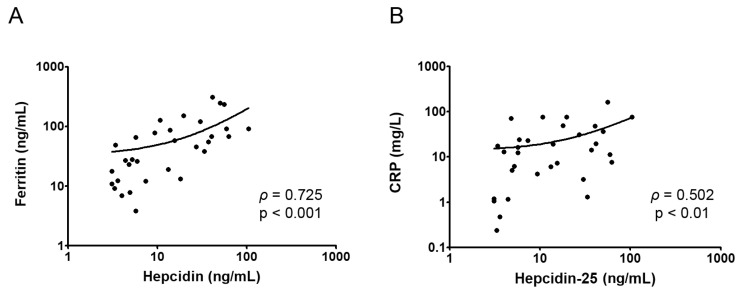
Correlations between hepcidin-25 and ferritin (**A**) and hepcidin-25 and CRP (**B**) in inflammatory bowel disease Group II.

**Figure 3 ijms-25-03564-f003:**
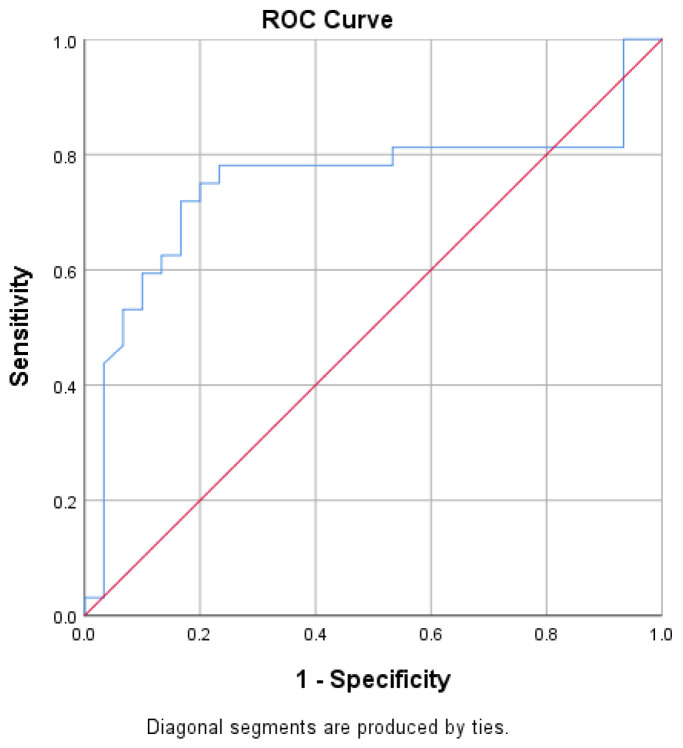
ROC curve analysis: at cut-off value of hepcidin-25 at 4.79 ng/mL, sensitivity is 78.1%, specificity 76.7%, AUC 0.749.

**Figure 4 ijms-25-03564-f004:**
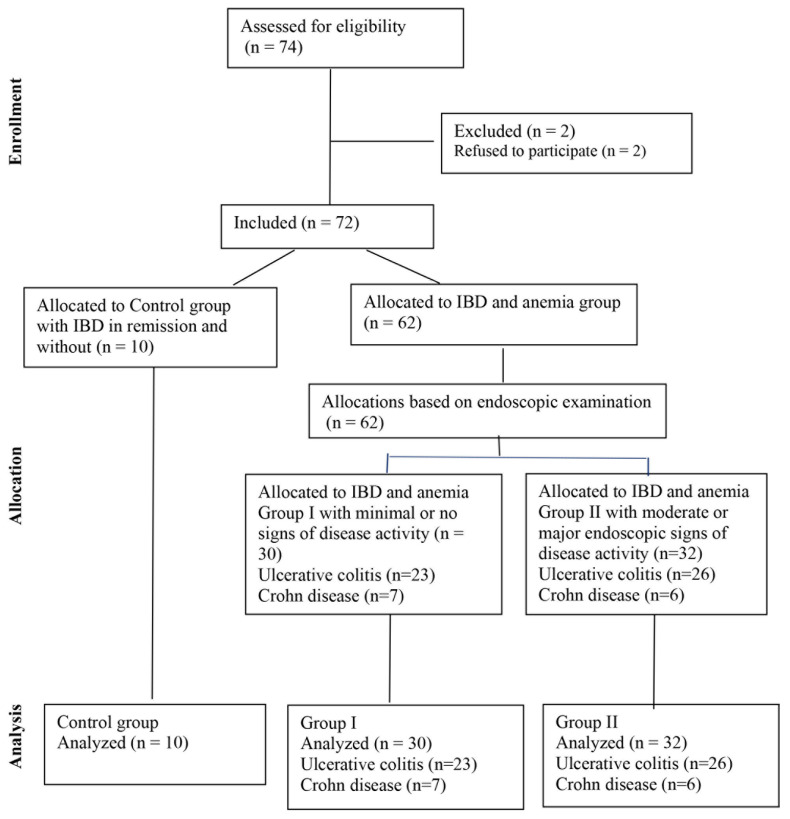
Study flowchart.

**Table 1 ijms-25-03564-t001:** Baseline clinical characteristics of the study population.

	Ulcerative Colitis	Crohn’ Disease	*p* Value *
Disease duration (year)	3.5 ± 0.3	5.1 ± 0.7	*p* < 0.001
Extent UC			
Left-sided colitis	65%	N/A	0.879
Extensive colitis	35%	N/A	
Location CD			
Ileal	N/A	46%	0.910
Colonic	N/A	23%	
Ileocolonic	N/A	31%	
Therapy			
Aminosalicylates (%)	20.41%	N/A	0.342
Systemic steroids (%)	6.12%	N/A	
Thiopurines (%)	10.20%	15.38%	
Anti-TNF (%)	36.74%	53.85%	
Vedolizumab (%)	26.53%	30.77%	
Endoscopic Mayo score			
Mayo 1 (%)	25.45%	N/A	0.976
Mayo 2 (%)	34.55%	N/A	
Mayo 3 (%)	40.00%	N/A	
Endoscopic SES-CD score			
Mild	N/A	30.35%	0.894
Moderate	N/A	54.50%	
Severe	N/A	15.15%	

Abbreviations: N/A, not applicable; UC, ulcerative colitis; CD, Chron’s disease; * For categorical values, we used a chi-square incorporating Yates’ correction for continuity (This correction was employed to improve the accuracy of the null-condition sampling distribution of chi-square).

**Table 2 ijms-25-03564-t002:** Linear regression analysis of variables influencing hepcidin-25 serum levels. Hepcidin-25 is a dependent variable. Multivariate analysis of Group I: F = 25.440, *p* < 0.001, R square 74.6%. Multivariate analysis of Group II: F = 4.626, *p* = 0.009, R square 33.1%.

	Group I				Group II			
	Univariate Analysis		Multivariate Analysis		Univariate Analysis		Multivariate Analysis	
Independent Variables	Beta	*p* Value	Beta	*p* Value	Beta	*p* Value	Beta	*p* Value
Ferritin	0.842	<0.001	0.904	<0.001	0.521	0.002	0.386	0.041
CRP	0.045	0.812	−0.202	0.062	0.406	0.021	0.243	0.189
Age	−0.236	0.21	N/A	N/A	0.002	0.99	N/A	N/A
Iron	0.002	0.991	N/A	N/A	0.074	0.688	N/A	N/A
Folic acid	−0.198	0.295	N/A	N/A	0.074	0.688	N/A	N/A
Hgb	0.386	0.035	−0.015	0.891	0.182	0.32	0.175	0.281
IL-6	0.065	0.734	N/A	N/A	−0.1	0.587	N/A	N/A

Abbreviations: N/A, not applicable; CRP, C-reactive protein; Hgb, hemoglobin.

## Data Availability

All data generated and/or analyzed during the current study are available from the corresponding author on reasonable request.

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
