# Peer review of "A Prospective Observational Study Analyzing the Diagnostic Value of Hepcidin-25 for Anemia in Patients with Inflammatory Bowel Diseases"

_ijms, 2024, doi:10.3390/ijms25073564_

Round 1
Reviewer 1 Report
Comments and Suggestions for Authors
Serbian authors analyzed the importance of hepcidin in IBD, but the manuscript requires significant rewriting. Below are my comments on the manuscript:
1. please correct the title of the manuscript, there is a slight error,
2. please provide the authors' institutional e-mail addresses, private ones are probably not accepted,
3. in the introduction, the authors should be sure that they have provided a reference source for each piece of information,
4. L62 - please write the gene name in italics,
5. in the introduction, there is a disproportion between the description of hepcidin and the description of the relationship between hemostasis and the pathophysiology of IBD or between hepcidin and hemostasis. In this form, the reader does not know what the authors' grounds were for conducting such an analysis.
6. Table 1 - please provide numbers for subgroups and p values for each comparison.
7. The results should be divided into smaller subsections to make it easier for the reader to understand the topic.
8. The results of hepcidin concentration tests are surprisingly low and unacceptable.
9. What is the number of patient subgroups within subgroups? Was the statistical analysis sensible there?
10. Correlations are widely known and do not contribute much to understanding the topic. Why were no such correlations found in group I?
11. I also don't see anything innovative in the results from Table 2 and Figure 4.
12. There is little emphasis in the results on the relationship between hemostasis and hepcidin. So how did the authors achieve this goal?
13. The discussion is too long. The authors did not discuss the results as a whole, but only analyzed their observations fragmentarily.
14. The literature must be expanded to include the following items:
10.3390/cancers16020332.
doi: 10.3390/ijms23073612.
15. More date on ELISA kits are required.
16. Please give inclusion and exclusion criteria.
Comments on the Quality of English LanguageModerate editing of English language required.
Author Response
Dear reviewer,
In the attachment is a file of all the answers to your kind suggestions.
Best regards

Reviewer 2 Report
Comments and Suggestions for Authors
Interesting study with interesting outcomes noted, through an observational design.
However I have a number of questions that merit clarification:
First of all: functional vs. absolute iron deficiency vs. anemia of chronic disease; where do you draw the line? The arbitary concentration of ferritin conc at 30 ng/ml is reflective of sideropenia (otherwise described as absolute deficiency), however the european society of gastroenterology guidelines do state that patients even with a ferritin of 100 ng/ml can be iron deficient. A lot of it in terms of iron deficiency is actually relevant to iron mobilisation and not just about absorption.
In line 59 you state: Multiple laboratory tests are used to identify cause of anemia. In my understanding estimation of anemia is done based on Hb; the other stated tests are performed to identify the cause of anemia. Please clarify.
In the exclusion criteria you state that eGFR < 30 is end stage renal disease. eGFR < 30 is not end stage renal disease.
You state in your exclusion criteria that previous treatment with iron was one. How long for, and with administration at what time point?
I struggle to see the originality within the manuscript. Inflammatory bowel disease is indeed a state of high inflammation, and we are already aware that inflammation is a driver of increased hepcidin. Are you suggesting of employing hepcidin as an inflammatory marker (it can be seen as surrogate, in combination with ferritin)? Are you also suggesting not treating someone with iron deficiency until inflammation settles? Also: at what point do we believe that oral iron may represent a better option than intravenous iron? Intravenous iron bypasses one of the two important hepcidin barriers - oral iron bypasses none. Please clarify these points further.
I see that anti-inflammatory treatment took place with modulators and biologics - was there any association with those and hepcidin/ferritin/anemia?
Did you correlate these findings to iron concentration transferrin saturation, sTFR/Log ferritin?
Overall, I need to congratulate you for your efforts and you well thought off plan and execution, however I believe that certain modifications need to take place in order to be able to convey a message to the reader.
Round 2
Reviewer 1 Report
Comments and Suggestions for Authors
The authors have satisfactorily addressed all of my concerns.